# The Effectiveness and Safety of Mind-Body Modalities for Mental Health of Nurses in Hospital Setting: A Systematic Review

**DOI:** 10.3390/ijerph18168855

**Published:** 2021-08-23

**Authors:** Su-Eun Jung, Da-Jung Ha, Jung-Hyun Park, Boram Lee, Myo-Sung Kim, Kyo-Lin Sim, Yung-Hyun Choi, Chan-Young Kwon

**Affiliations:** 1Department of Clinical Korean Medicine, Graduate School, Dong-eui University, Busan 47227, Korea; jungse0906@gmail.com (S.-E.J.); ogjen6019@naver.com (D.-J.H.); jhpark6769@gmail.com (J.-H.P.); 2Clinical Research Coordinating Team, Korea Institute of Oriental Medicine, Daejeon 34054, Korea; qhfka9357@naver.com; 3Department of Nursing, College of Nursing, Healthcare Sciences & Human Ecology, Dong-eui University, Busan 47340, Korea; myosg@deu.ac.kr; 4Department of Music, Graduate School, Pyeongtaek University, Pyeongtaeksi 17869, Gyeonggi-do, Korea; shimkl@naver.com; 5Department of Biochemistry, College of Korean Medicine, Dong-eui University, Busan 47227, Korea; choiyh@deu.ac.kr; 6Department of Oriental Neuropsychiatry, College of Korean Medicine, Dong-eui University, Busan 47227, Korea

**Keywords:** nurse, mental health, burnout, mind-body medicines, systematic review

## Abstract

The mental health of nurses including burnout is an important issue. The purpose of this systematic review was to evaluate whether mind-body modalities improve burnout and other mental health aspects of nurses. A comprehensive search was conducted using six electronic databases. Randomized controlled trials using mind-body modalities on the mental health of nurses, up to January 2021, were included. The methodological quality of the included studies was assessed using the Cochrane Risk of Bias tool. Seventeen studies were included in the review. Data on mindfulness-based interventions (MBIs) and yoga were available for burnout, and there was no evidence that multimodal resilience programs including MBIs statistically significantly improved burnout levels compared to no intervention or active control groups. However, one study reported that yoga could significantly improve emotional exhaustion and depersonalization, which are subscales of burnout, compared to usual care. In addition, the effects of MBIs, relaxation, yoga, and music on various mental health outcomes and stress-related symptoms have been reported. In conclusion, there was some evidence that yoga was helpful for improvement in burnout of nurses. However, due to the heterogeneity of interventions and outcomes of the studies included, further high-quality clinical trials are needed on this topic in the future.

## 1. Introduction

The mental health problems of nurses are common, and nurses are exposed to a variety of mental health risk factors, including work demands, psychological demands, violence, aggression, poor relationships with administrators, accidents involving the risk of exposure to human immunodeficiency virus, stress, and errors in the execution of labor activities [1]. Among various mental health problems, burnout is a major cause of emotional exhaustion, high depersonalization, and low personal accomplishment, and its prevalence in primary care nurses is common at 28%, 15%, and 31%, respectively [2]. Importantly, the burnout of nurses can affect, beyond the individual level, the organization to which they belong, or patient outcomes on account of their decreased job performance, more sick leave, and more absences [3].

Mind-body modalities have been used to cope with stress-related problems for a long time, and as evaluated currently, they are moving to the mainstream based on the empirical clinical evidence accumulated in the aspect of evidence-based medicine (EBM) [4]. Mind-body modalities including tai chi, qigong, yoga, and meditation have been used in several populations to manage stress and improve physical and mental health, including in patients with cardiac disease [5], multiple sclerosis [6], fibromyalgia [7], and chronic pain [8]. Moreover, according to the 2007 National Health Interview Survey, adults with common neurological conditions, such as headache, stroke, and cognitive impairment, used mind-body modalities significantly more often than adults who did not, at a rate of 24.5% [9]. Studies examining the biological mechanisms of mind-body modalities have confirmed that these treatments have immunomodulatory and anti-inflammatory properties [10,11] that may be related to enhanced neurogenesis and neuroplasticity [12]. In addition, studies have reported that mind-body modalities may have a positive role in stress responsiveness by affecting biomarkers such as cortisol, dehydroepiandrosterone sulfate, and testosterone accordingly [13].

Likewise, mind-body modalities have the potential to improve nurses’ overall health and level of well-being and prevent and/or reduce burnout levels by alleviating the accompanying physical symptoms as well as improving the psychological stress of nurses. For example, a tertiary care hospital in the United States reported that mindfulness-based stress reduction (MBSR), a typical type of mindfulness-based interventions (MBIs), was introduced to staff nurses in the hospital, and it was reported that the program effectively reduced job burnout and improved mindfulness, self-compassion, and serenity in the participants [14]. Mindfulness practice is also being considered as an effective self-management method or a stress reduction technique for healthcare workers including nurse exposed to the COVID-19 pandemic and threatened with their mental health and well-being [15,16]. Therefore, examining the impact of mind-body modalities on the mental health of nurses will potentially help establish strategies to improve the mental health of healthcare workers in this unprecedented pandemic, and further potentially increase humanity’s capacity to respond to this pandemic. However, no study has systematically analyzed the effectiveness of mind-body modalities on the mental health problems of nurses. The purpose of this systematic review was hence to evaluate whether mind-body modalities improve burnout and other mental health aspects of nurses in hospital setting.

## 2. Methods

This systematic review complied with the Preferred Reporting Items for Systematic Reviews and Meta-Analyses (PRISMA) statement for reporting systematic reviews and meta-analyses [17] (Appendix A). The protocol of this systematic review was registered in OSF registries (doi:10.17605/OSF.IO/U8P3T), and this review followed the protocol.

### 2.1. Study Search

Comprehensive searches were conducted in a total of six international electronic databases, including MEDLINE (via PubMed), EMBASE (via Elsevier), the Cochrane Library Central Register of Controlled Trials, the Cumulative Index of Nursing and Allied Health Literature, the Allied and Complementary Medicine Database, and PsycARTICLES. In addition, a manual search on Google Scholar was conducted to search for gray and potentially missing literature, and a list of references from related papers including the studies included in this review was reviewed accordingly. The search date was 28 January 2021, and all related studies published up to the search date were reviewed. The study search was performed by a single researcher (Lee B). Search strategies for each database and search results are presented in Appendix A.

### 2.2. Study Selection and Inclusion Criteria

Two independent researchers (Jung SE and Ha DJ) selected the studies according to the following inclusion criteria: Title/abstract and full-text review. In case of disagreement between them, a third-party researcher (Kwon CY) intervened accordingly. (1) Population: Nurses in hospital settings, regardless of sex, age, and ethnicity (2) Intervention: Mind-body modalities including meditation, mindfulness intervention, autogenic training, yoga, tai chi, qigong, breathing exercise, music therapy, guided imagery, and biofeedback. (3) Comparator: No treatment, wait-list, sham control, attention control, or active comparators. (4) Outcomes: The primary outcome included the level of burnout assessed using validated assessment tools such as the Maslach Burnout Inventory [18], and secondary outcomes included all other mental health aspects or stress-related symptoms. (5) Study design: Only randomized controlled trials (RCTs) were allowed, and no restrictions on language were imposed in the study. That is, there are no restrictions on language sources in the selection for review.

### 2.3. Risk of Bias Assessment

The methodological quality of the included studies was assessed by two independent reviewers (Jung SE and Ha DJ) using the Cochrane risk of bias (RoB) tool. This tool judges the RoB of RCTs as high, low, or unclear in the domains of selection, performance, detection, attrition, reporting, and other biases. RoB evaluation followed the method described in the Cochrane Handbook for Systematic Reviews of Interventions version 5.1.0 [19]. In case of disagreement between the two independent reviewers, a third-party researcher (Kwon CY) intervened accordingly. The evaluated RoB results are presented as figures using RevMan 5.4. 

### 2.4. Data Extraction

The following data from the eligible studies were extracted and entered into a Microsoft Excel file: Publication year, name of 1st author, information for RoB assessment, country where the study was performed, sample size, mean age, ward in which the participants worked, pathological condition of participants, treatment intervention, control intervention, intervention period, time of assessment, outcomes, and results.

### 2.5. Data Analysis

All included studies were analyzed qualitatively. Quantitative synthesis was not performed considering the heterogeneity of the interventions used in the included studies. Instead, the reported results were classified into occupational and environmental outcomes, individual resistance to stress, global health and wellness, psychological symptoms, physical symptoms, and biological data according to their characteristics. In addition, interventions were classified into MBIs, relaxation, yoga, music, and aromatherapy according to their characteristics. Upon comparison of the two groups, a case with *p*-value less than 0.05 was considered to be statistically significant.

### 2.6. Publication Bias

Due to the heterogeneity of the interventions and outcomes of the included studies, quantitative synthesis was not performed in this study. Therefore, publication bias using a funnel plot could not be evaluated.

## 3. Results

### 3.1. Study Search

As a result of the literature search, 16,129 documents were identified after excluding duplicates. Among them, 39 potentially relevant articles were selected using the first title/abstract screening process. As a result of secondary screening for the full texts, 13 that were not RCTs [20,21,22,23,24,25,26,27,28,29,30,31,32], three that did not use mind-body modalities [33,34,35], four which were without details (conference abstract) [36,37,38,39], and two using the same data as other journal articles (thesis or conference abstract) [40,41], were excluded from the study. Finally, a total of 17 RCTs were included in this review [42,43,44,45,46,47,48,49,50,51,52,53,54,55,56,57,58] (Figure 1).

### 3.2. Characteristics of Included Studies

Among the included studies, 15 were parallel RCTs [42,43,45,46,47,48,49,50,51,53,54,55,56,57,58] while the other two were cross-over RCTs [44,52]. All these studies were published between 1993 and 2020. Three studies each were conducted in the US [45,46,57] and China [47,50,51], two each in Taiwan [42,44] and Japan [52,55], and one each in Hong Kong [43], Korea [48], Greece [49], Turkey [53], France [54], Malaysia [56], and Iran [58]. The analyzed sample sizes of the included studies varied from 27 to 224. Four studies used yoga [46,47,52,54] as their intervention, three used music [44,49,58], and one used music combined with aromatherapy as well as music in its four-arm RCT [58]. Seven studies used MBIs, including MBSR in three [45,50,51], meditation [48], and mindfulness-based programs in three [55,56,57]. Three studies used relaxation [42,43,53]. Two studies used multimodal interventions, such as multimodal resilience training programs, including MBSR [45] and community resiliency models, including mindful eating [57] (Table 1).

### 3.3. Methodological Qualities of Included Studies

In the random sequence generation domain, 10 studies [42,47,48,50,51,53,54,55,56,58] that used proper random sequence generation methods such as random number tables or simple randomization were rated as having low RoB, while the other seven [43,44,45,46,49,52,57] without a description of randomization method were rated to have unclear RoB. None of the included studies described allocation concealment and blinding of participants and personnel. In the blinding of participants and personnel domain, three [53,56,58] were rated as having a high RoB due to the nature of the intervention, while the others [42,43,44,45,46,47,48,49,50,51,52,54,55,57] were rated as having an unclear RoB. Only two studies [55,58] described that they performed the blinding of outcome assessment, while the rest of the studies [42,43,44,45,46,47,48,49,50,51,52,53,54,56,57] did not. In eight studies, there were no drop-out cases [43,44,45,46,49,52,54,58]. Some drop-out cases existed in the remaining studies; however, the numbers of the cases probably did not affect the study results [42], they were similar between groups with appropriate reasons for drop-out [48,51,53], and/or appropriate statistical analysis (i.e., intent-to-treat analysis) was applied accordingly [55,56]. Since no previously published protocols were identified except for one study [55], the other studies [42,43,44,45,46,47,48,49,50,51,52,53,54,56,57,58] were evaluated as having unclear RoB in the selective reporting domain. Regarding other biases, 13 studies [43,44,46,47,48,50,51,52,53,54,55,57,58] describing statistically homogeneous demographic and clinical characteristics between the groups at baseline were evaluated as having low RoB. One study [56] describing the significant difference between the groups at baseline was evaluated as having a high RoB. The other three did not describe statistical homogeneity between the groups at baseline [42,45,49] (Figure 2 and Figure 3).

### 3.4. Main Results

#### 3.4.1. Primary Outcome (Burnout)

(1) MBIs: Two studies using a multimodal resilience program including mindfulness training were compared with no intervention or nutrition/healthy eating group. In the former [45], no statistical comparison was performed between groups using the Maslach Burnout Inventory, and in the latter [57], no statistically significant difference was found between the two groups in the Copenhagen Burnout Inventory *(p* = 0.777). Watanabe (2019) found that there was no significant difference between the brief mindfulness-based stress management program group and the psychoeducation group in any subscale of the Maslach Burnout Inventory *(p* = 0.266 to 0.664) [55] (Table 2).

(2) Yoga: Alexander (2015) reported that yoga had a statistically superior improvement in emotional exhaustion *(p* = 0.041) and depersonalization (*p* = 0.035), but not in the lack of personal accomplishment (*p* = 0.554), among the Maslach Burnout Inventory compared to usual care [46] (Table 2).

#### 3.4.2. Secondary Outcomes

Except for the case of not performing a statistical comparison between the two groups, it is as follows:

(1) MBIs: Regarding the outcomes of occupation and environment, compared to no intervention, Ghawadra (2020) [56] found that MBIs significantly improved job satisfaction as assessed by the Job Satisfaction Scale (*p* = 0.040) compared to no intervention, while Lin (2019) [51] found that MBSR did not significantly improve job satisfaction as assessed by the McCloskey/Mueller Satisfaction Scale (*p* > 0.05). Additionally, according to Watanabe (2019) [55], there was no significant difference in absolute presenteeism (*p* = 0.258) and relative presenteeism (*p* = 0.065) in the World Health Organization Health and Work Performance Questionnaire between MBIs and psychoeducation groups. Regarding the outcomes of the individual’s resistance to stress, Lin (2019) [51] found that MBIs did not significantly improve resilience assessed by the Connor–Davidson Resilience Scale (CDRS) (*p* > 0.05) compared to no intervention, and Grabbe (2020) [57] reported that there was no significant difference in CDRS (*p* = 0.910), between multimodal intervention including mindful eating group and nutrition/healthy eating group. Chang (2016) [48] found that, compared to no intervention, meditation significantly improved the sense of power assessed by Power as Knowing Participation in Change Tool (*p* = 0.001 to 0.049 according to the subscale). According to Ghawadra (2020) [56], there was no significant difference in absolute mindfulness assessed by the Mindful Attention Awareness Scale (*p* = 0.967) between the MBI and no intervention groups. Regarding the outcomes of the individual’s global health and wellness, Chang (2016) [48] found that, compared to no intervention, meditation significantly improved global score and most subscales of quality of life assessed using the abbreviated World Health Organization Quality of Life (WHOQOL-BREF) questionnaire (*p* = 0.006 to 0.039 for global score, physical domain, psychological domain, and social domain; *p* = 0.057 for environmental domain), while Watanabe (2019) [55] found no significant difference in quality of life assessed by the EuroQol five-dimension scale utility score between the MBIs group and the psychoeducation group (*p* = 0.131). Grabbe (2020) [57] found that there was no significant difference in well-being assessed by the five-item World Health Organization Well-Being Index (*p* = 0.168), between the multimodal intervention including mindful eating group and nutrition/healthy eating group. Regarding the outcomes of psychological symptoms, Yang (2018) [50] found that MBSR significantly improved psychological pathology assessed by the Symptom Checklist-90-Revised (*p* < 0.001), stress level assessed by the Nursing Stress Scale (*p* < 0.001), anxiety level assessed by Self-Rating Anxiety Scale (*p* < 0.001), and depression level assessed using the Self-Rating Depression Scale (*p* < 0.001), compared to routine psychological support and activities. Lin (2019) [51] also found that MBSR significantly improved stress levels assessed by the Perceived Stress Scale (PSS) (*p* < 0.01) and positive and negative emotions assessed by the Positive and Negative Affect Schedule (*p* < 0.01 to 0.05, according to the subscale). However, in Grabbe’s study (2020) [57], there was no significant difference in posttraumatic stress assessed by the secondary traumatic stress scale (*p* = 0.846) between the multimodal intervention including mindful eating and in the nutrition/healthy eating groups. Moreover, Watanabe (2019) [55] found that there was no significant difference in depression (*p* = 0.192) and anxiety (*p* = 0.190) assessed by the Hospital Anxiety and Depression Scale (HADS), anxiety assessed by the Generalized Anxiety Disorder seven-Item Scale (*p* = 0.057), depression assessed by the Patient Health Questionnaire-9 (*p* = 0.315), and insomnia assessed using the Insomnia Severity Index (ISI) (*p* = 0.435), between the MBIs and psychoeducation groups. In the case of Ghawadra (2020) [56], the MBI group showed significantly improved scores for anxiety (*p* = 0.008), but not for stress (*p* = 0.159) and depression (*p* = 0.709) in the Depression, Anxiety, and Stress Scales-21, compared to the no-intervention group. Regarding the outcomes of somatic symptoms, Grabbe (2020) [57] reported that there was no significant difference in somatic symptoms assessed using the Somatic Symptoms Scale-8 (*p* = 0.563), between the multimodal intervention including mindful eating group and nutrition/healthy eating group (Table 2).

(2) Relaxation: Regarding outcomes on the individual’s resistance to stress, according to Ozgundondu (2019) [53], progressive muscle relaxation (PMR) group showed significantly improved score of acceptance (*p* = 0.038), but not of use of instrumental support (*p* = 0.980), humor (*p* = 0.425), active coping (*p* = 0.237), substance use (*p* = 0.631), venting (*p* = 0.235), religion (*p* = 0.108), denial (*p* = 0.302), behavioral disengagement (*p* = 0.413), self-distraction (*p* = 0.224), self-blame (*p* = 0.758), positive reframing (*p* = 0.295), use of emotional support (*p* = 0.101), and planning (*p* = 0.160), in the abbreviated version of the Coping Orientation to Problems Experienced Inventory, compared to the education group. Regarding the outcomes of an individual’s global health and wellness, Tsai (1993) [42] reported that relaxation significantly improved general health as assessed by the General Health Questionnaire (GHQ) (*p* < 0.05), compared to traditional in-service education. However, in Yung’s study (2004) [43], there was no significant difference in GHQ between the stretch-release relaxation group, the cognitive relaxation group, and the no-intervention group (*p* = 0.320). Regarding the outcomes of psychological symptoms, Tsai (1993) [42] reported that relaxation significantly improved stress levels assessed by the Nurse Stress Checklist (*p* < 0.05), compared to a traditional in-service education. In addition, Ozgundondu (2019) [53] reported that the PMR group showed significantly better results on PSS (*p* = 0.030) than the education group. However, in Yung’s study (2004) [43], there was no significant difference in anxiety state (*p* = 0.097) and anxiety trait (*p* = 0.679) in the Stat-Trait Anxiety Inventory, between the stretch-release relaxation group, the cognitive relaxation group, and the no-intervention group. 

Regarding the outcomes of somatic symptoms, Ozgundondu (2019) [53] found that PMR significantly improved fatigue levels assessed by the Fatigue Severity Scale (*p* < 0.001), compared to education (Table 2).

(3) Yoga: Regarding the outcomes of the individual’s resistance to stress, Alexander (2015) [46] found that the yoga group showed significantly better results on healthy lifestyle assessed by Health Promoting Lifestyle Profile II (*p* = 0.006), but not on the mindfulness level assessed by the Freiburg Mindfulness Inventory (*p* = 0.067), compared to usual care. Regarding the outcomes of individual’s global health and wellness, Rostami (2019) [54] found that all subscales in the WHOQOL-BREF, including global score and physical, psychological, social, and environmental domains (all, *p* < 0.001) were significantly better in the yoga group than in the no-intervention group. Regarding the outcomes of psychological symptoms, Miyoshi (2019) [52] found that the yoga group showed significantly better results on stress level assessed by the Brief Job Stress Questionnaire (*p* = 0.01), compared to the stress relief method group. Fang (2015) [47] found that the yoga group showed a significantly lower number of scores more than 32 (which means high stress level) (*p* = 0.001) on the Questionnaire on Medical Workers’ Stress, compared to the no-intervention group (Table 2).

(4) Music: Regarding the outcomes of psychological symptoms, Lai (2011) [44] found that music significantly improved stress levels assessed by self-perceived stress (*p* < 0.001), compared to chair rest. However, in Ploukou’s study (2018) [49], there was no significant difference in depression scores in the HADS (*p* > 0.05) between the percussion music group and the no-intervention group. In Zamanifar’s study (2020) [58], the music therapy group (*p* = 0.0001), the aromatherapy group (*p* = 0.0001), and the music therapy combined with aromatherapy group (*p* = 0.0001) showed significantly lower Beck Anxiety Inventory scores than the no-intervention group. Regarding the outcomes of somatic symptoms, Ploukou (2018) [49] did not find significant differences in psychosomatic symptoms assessed by the Pennebaker Inventory of Limbic Languidness (*p* > 0.05), between the percussion music group and the no-intervention group. Regarding the outcomes on biological data, Lai (2011) [44] found that the music group showed significantly less mean arterial blood pressure (BP) (*p* < 0.001), serum cortisol (*p* < 0.025), and heart rate (HR) (*p* < 0.001), and significantly higher finger temperature (*p* < 0.001), compared to the chair rest group (Table 2).

#### 3.4.3. Safety Data

Only three studies reported safety data [52,53,55], and no adverse events occurred in these studies.

## 4. Discussion

### 4.1. Main Findings

This review aimed to comprehensively analyze the effectiveness and safety of mind-body modalities on burnout and other mental health aspects of nurses in hospital settings. A total of 17 RCTs were included, and the mind-body modalities used were classified into MBIs, relaxation, yoga, and music according to their characteristics.

Data on MBIs and yoga were available for burnout, and there was no evidence that multimodal resilience programs including MBSR or mindful eating and MBIs statistically significantly improved burnout levels compared to the no-intervention group or active control groups. However, one study [46] reported that yoga could significantly improve emotional exhaustion and depersonalization, which are subscales of burnout, compared to usual care.

The effects of MBIs, relaxation, yoga, and music on several secondary outcomes, including the primary outcome, can be summarized as follow. In the case of MBIs, studies have reported that MBIs showed significant benefits on the sense of power, psychological pathology, positive and negative emotions, but showed non-significant effects on presenteeism, resilience, mindfulness, well-being, posttraumatic stress, insomnia, and somatic symptoms, compared to control groups. Mixed results have been reported for job satisfaction, quality of life, stress, anxiety, and depression. In the case of relaxation, studies have reported that relaxation has significant benefits on coping style, stress, and fatigue, but a non-significant effect on anxiety, compared to control groups. Mixed results have been reported for general health. In the case of yoga, studies have reported that it showed significant benefits on healthy lifestyle, quality of life, and stress, but a non-significant effect on mindfulness, compared to control groups. In the case of music, studies have reported that it showed significant benefits on stress, anxiety, and some biological data, including mean arterial BP, serum cortisol, HR, and finger temperature, but non-significant effects on depression and somatic symptoms, compared to control groups, although it was the immediate result of one session in the study. Overall, stress could be positively affected by most mind-body modalities. However, the methodological quality of the included studies was not optimal. This means that the certainty of the findings of this review is not high, suggesting that our findings may change depending on the results of future high-quality and large-scale studies.

### 4.2. Clinical Implications

The mental health of nurses is both individually and socially an important issue [3]. Mind-body modalities are promising for managing a variety of physical and psychiatric symptoms, including anxiety, depression, perceived stress, coronary artery disease, headaches, insomnia, incontinence, chronic lower back pain, and even cancer symptoms [4]. In this review, we examined the potential applications of several mind-body modalities in terms of improving the mental health of nurses. In particular, in this review, the authors conducted a comprehensive study search and data analysis to establish clinical evidence for reliable data of mind-body modalities from the perspective of EBM. However, the findings were limited by the consequent lack of the number of included studies and the lack of methodological quality of the included studies. Although mind-body modalities may have beneficial impacts on improving mental health, mental health outcomes in this population vary widely, and the mind-body modalities used in the studies included in this review also varied. Meanwhile, studies evaluating the effectiveness of the potentially most effective mind-body modalities as well as the most treatment-sensitive outcomes may have clinical significance. In that sense, this review provides comprehensive evidence on this topic by performing a comprehensive analysis of the available studies. However, this review also points out that there are still many areas that have not been studied, such as the heterogeneity of different interventions or outcomes.

### 4.3. Limitations and Suggestions for Further Studies

This review presents results of the most comprehensive systematic review available on mind-body modalities for burnout as well as other mental health aspects of nurses. The use of various mind-body modalities as well as various mental-health-related outcomes has been identified for this issue, and its heterogeneity is also highlighted in this review. The following limitations should be considered when interpreting this study.

First, the number of studies included in this review was small, and the interventions and outcomes used were also varied. As a result, the results for each outcome depended on the results from one to three studies in general, and the evidence supported by the small number of studies with small sample sizes cannot be evaluated as having sufficient reliability. Therefore, further research on this topic is required. Second, the overall methodological quality of the included studies was poor. In particular, there was ambiguity regarding the blinding procedure in most included studies, potentially suggesting that the results reported in these studies may have been overestimated or influenced by the placebo effect. Therefore, it is necessary to conduct a high-quality clinical trial with proper blinding procedures in relation to this topic in the future. Third, in this review, we categorized mind-body modalities into four categories, but the details may vary for each mind-body modality. For example, one three-arm study [43] used both stretch-release relaxation and cognitive relaxation, which were both considered as relaxations by us. In addition, even with the same MBSR, in a standardized mindfulness program, there are some modifications depending on the characteristics of the participants or the environment. That is, the classification of this review does not sufficiently guarantee the homogeneity of the individual interventions analyzed. In future studies, based on the findings of this review, it may be possible to specify population, intervention, comparator, and outcomes by focusing on mental health-related outcomes where specific mind-body modalities can have positive effects. Fourth, the reported outcomes in the included studies were too diverse to be meta-analyzed. Unlike the clear primary outcome for patients with certain mental disorders, nurses are a non-clinical population, and it is thought that this is so because there is no gold standard primary outcome for nursing mental health. However, some outcomes such as job-related stress, job satisfaction, burnout, and the occurrence of medical errors may be unique and relevant in this population. Fifth, data on biology among the outcomes reported in the included studies were lacking, although one study [44] reported the effects of music on biomarkers including BP, serum cortisol, HR, and finger temperature. Reporting effects on these biomarkers, as well as participant self-reports, are necessary to objectively evaluate the effectiveness of the interventions studied. Currently, considerable research is being conducted on biomarkers related to mental health, including genetics, transcriptomics, proteomics, metabolomics, and epigenetics [59]. Therefore, evaluation of these mental-health-related biomarkers should be introduced in the future. Finally, because quantitative synthesis was not performed due to the heterogeneity between the studies included in this review, an evaluation of publication bias through funnel plot generation was not possible [60], but this does not prove the absence of publication bias. That is, the possibility of publication bias suggests that unpublished studies in this field may have exaggerated the net benefit of the intervention [61]. Therefore, the findings of this review should be interpreted with caution.

## 5. Conclusions

This review is the most comprehensive one available on the impact of mind-body modalities on burnout as well as other mental health aspects of nurses in hospital settings. For burnout, which was the primary outcome, there was some evidence that yoga was helpful for improvement. In addition, the outcome that reported improved results in all mind-body modalities, including MBIs, relaxation, yoga, and music, was perceived stress. However, due to the heterogeneity of interventions and outcomes, and the methodological limitations of the studies included, further high-quality clinical trials are needed on this topic in the future.

## Figures and Tables

**Figure 1 ijerph-18-08855-f001:**
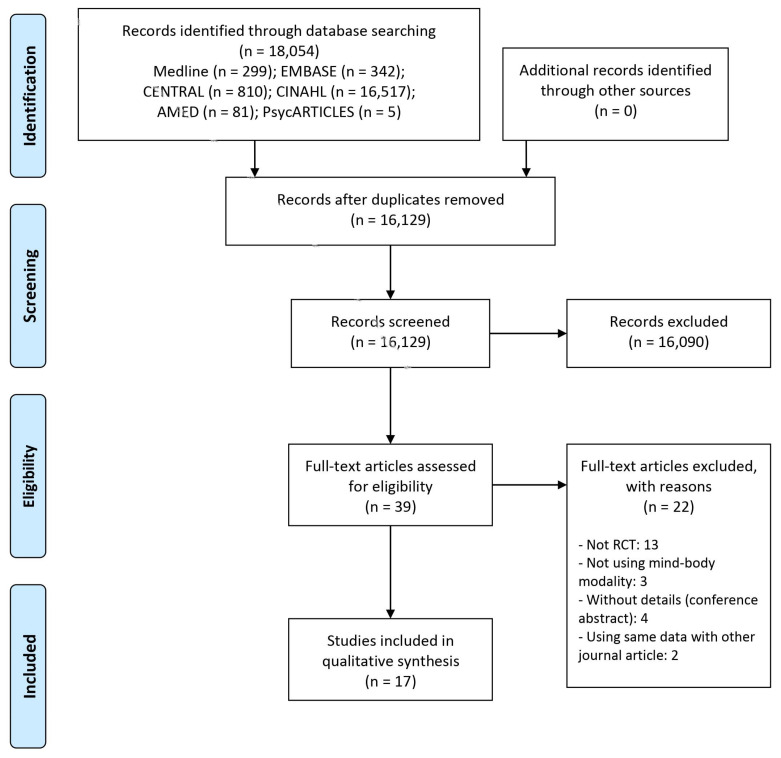
A PRISMA flow diagram of the literature screening and selection processes. AMED, Allied and Complementary Medicine Database; CENTRAL, Cochrane Central Register of Controlled Trials; CINAHL, Cumulative Index to Nursing and Allied Health Literature.

**Figure 2 ijerph-18-08855-f002:**
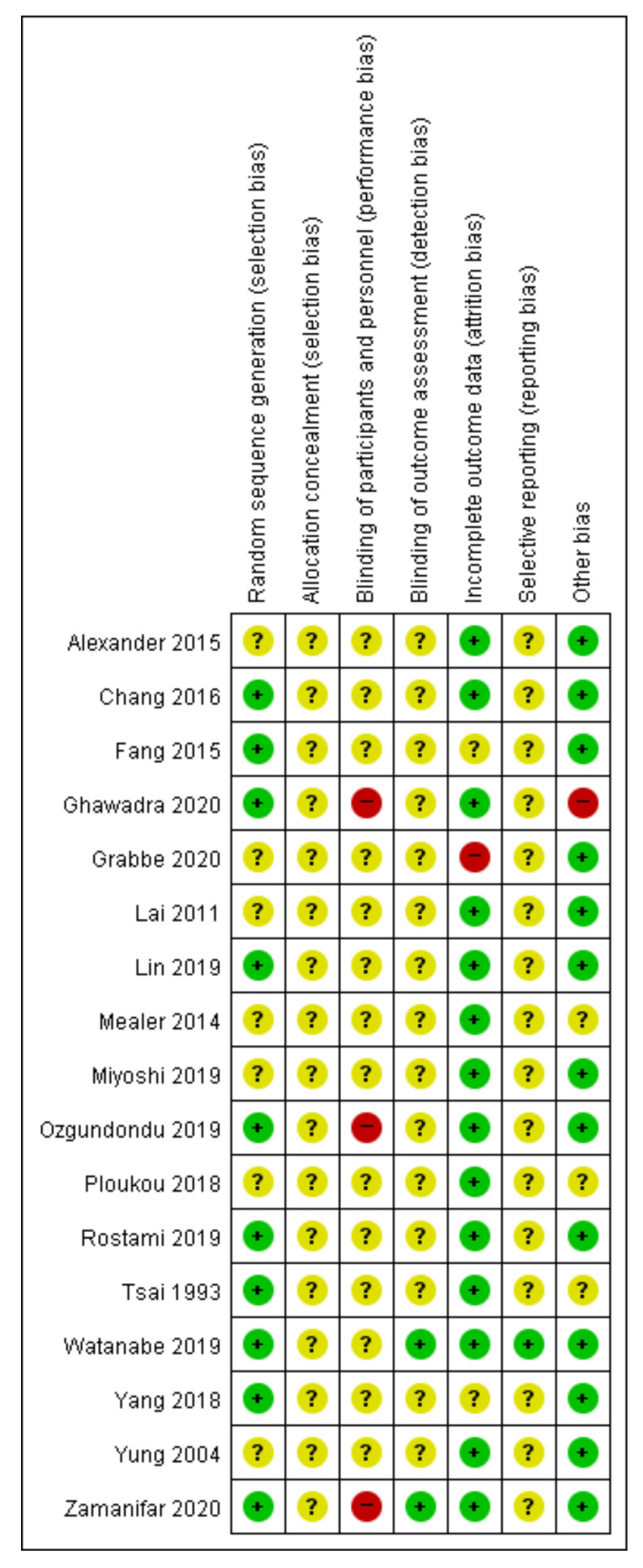
Risk of bias summary for all included studies. Low, unclear, and high risk, respectively, are represented with the following symbols: “+”, “?”, and “–”.

**Figure 3 ijerph-18-08855-f003:**
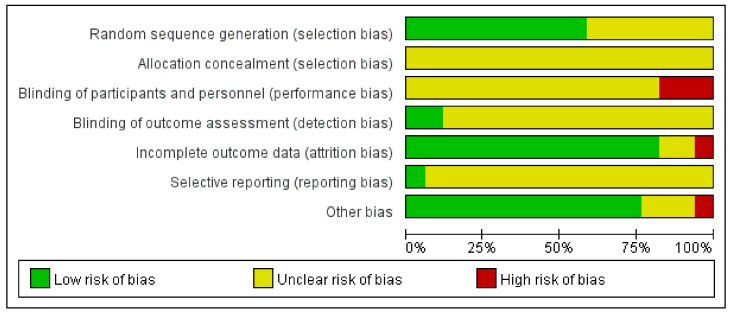
Risk of bias graph for all included studies.

**Table 1 ijerph-18-08855-t001:** Characteristics of included studies.

Study ID (Study Design/Country)	Sample Size (Included→Analyzed)	Mean Age (Range) (Years)	Ward In Which Nurses Work	Pathological Condition	Treatment Intervention	Control Intervention	Intervention Period (Assessment)	Outcomes
Tsai 1993 (Parallel RCT/Taiwan)	137→134 TG: unclear CG: unclear	25.2 (21–41)	MU, SU, ICU, PED, OB, OR, OPD, GYN, etc.	NA	Relaxation training (90 min/session, 1 session/week)	Traditional in-service education through lectures on theory analysis (90 min/session, 1 session/week)	2 weeks (Week 0, 2, 5)	1. NSC; 2. GHQ
Yung 2004 (Parallel RCT/Hong Kong)	65→65 TG1: 17→17 TG2: 18→18 CG: 30→30	NR	NR	NA	TG1: Stretch-release relaxation TG2: Cognitive relaxation (20 min/session, 1 session/week)	No intervention	4 weeks (Week 0, 4)	1. State-STAI; 2. Trait-STAI; 3. GHQ
Lai 2011 (Cross-over RCT/Taiwan)	54→54 TG: 27 (music→chair rest) CG: 54 (chair rest→music)	23.4 ± 2.46	MU, SU, maternal & PED, ICU	VAS stress ≥ 6	Music (30 min/session)	Chair rest (30 min/session)	1 x 1 session, wash-out 20 min (Min 0, 15, 30)	1. Self-perceived stress (0–10 cm VAS); 2. HR; 3. Mean arterial BP (mmHg); 4. Finger temperature; 5. Serum cortisol (ug/dL)
Mealer 2014 (Parallel RCT/America)	27→27 TG: 13 CG: 14	NR	ICU	82 ≥ CDRS	Multimodal resilience training program (2-day educational workshop; written exposure therapy (twelve 30 min sessions via email); MBSR (self-practice, 15 min at least 3 times per week); aerobic exercise (30–45 min at least 3 times per week); event-triggered counseling (30–60 min per session if needed))	No intervention	1. HADS; 2. MBI; 3. PDS; 4. CDRS	
Alexander 2015 (Parallel RCT/America)	40→40 TG: 20 CG: 20	NR	NR	NA	Yoga (1 session/week)	Usual care	8 weeks (Week 0, 8)	1. HPLP-II; 2. FMI; 3. MBI
Fang 2015 (Parallel RCT/China)	120→105 TG: 61→54 CG: 59→51	TG: 35.13 ± 10.98 CG: 36.05 ± 9.91 (25–51)	NR	NA	Yoga (50–60 min/session, more than 2 sessions/week)	No intervention	6 months (Month 0, 6)	1. PSQI; 2. QMWS
Chang 2016 (Parallel RCT/Korea)	50→40 TG: 25→21 CG: 25→19	31.5 ± 5.45 TG: 30.9 ± 5.59 CG: 32.1 ± 5.38	nurses of special nursing department	NA	Meditation (1–1.5 hours/session, 1 session/week)	No intervention	8 weeks (Week 0, 8)	1. PKPCT v II; 2. WHOQOL-BREF
Ploukou 2018 (Parallel RCT/Greece)	48→48 TG: 22 CG: 26	NR	ONC	NA	Percussion music (1 h/session, 1 session/week)	No intervention	4 weeks (Week 0, 4)	1. HADS; 2. PILL
Yang 2018 (Parallel RCT/China)	100→95 TG: 50→48 CG: 50→47	29.5 ± 7.1	PSY	positive in more than 30 items of SCL-90-R	MBSR (1 session/week)	Routine psychological support and activities.	8 weeks (Week 0, 8)	1. SCL-90-R; 2. SDS; 3. SAS; 4. NSS
Lin 2019 (Parallel RCT/China)	110→90 TG: 55→44 CG: 55→46	TG: 32.86 ± 7.49 CG: 30.20 ± 6.09	NR	NA	MBSR (2.5 h/session, 1 session/week)	No intervention	8 weeks (Week 0, 8, 12)	1. PSS; 2. PANAS; 3. CDRS; 4. MMSS
Miyoshi 2019 (Cross-over RCT/Japan)	20→20 TG: 10 (yoga→normal stress relief) CG: 10 (normal stress relief →yoga)	28.7 ± 4.9 (24–39)	NR	NA	Yoga (initial 1 h group session, and then self-practice, more than 5–15 min/session, more than 3 times/week)	Stress relief (including sleeping, shopping, and chatting with friends)	4 weeks × 4 weeks, wash-out 1 week (Week 0, 5, 10)	1. BJSQ
Ozgundondu 2019 (Parallel RCT/Turkey)	63→56 TG: 31→28 CG: 21→28	TG: 24.61 ± 2.61 CG: 27.75 ± 4.75	IM, ANE, CICU	NA	PMR (20 min/session, 1 session/week)	Face-to-face attention-matched education (a 20 min session)	8 weeks (Week 0, 4, 8, 12)	1. PSS; 2. FSS; 3. Brief COPE
Rostami 2019 (Parallel RCT/France)	70→70 TG: 35 CG: 35	TG: 30.5 ± 5.14 CG: 29.3 ± 5.1	ICU	NA	Yoga (2 session/week)	No intervention	8 weeks (Week 0, 4, 8, 24)	WHOQOL–BREF
Watanabe 2019 (Parallel RCT/Japan)	80→76,75 TG: 40→37 (assessed via phone), 36 (via internet) CG: 40→39 (via both phone and internet)	TG: 30.2 ± 9.0 (21–53) CG: 30.0 ± 7.9 (22–55)	IPD	NA	Brief mindfulness-based stress management program (30 min/session, 4 session/week)	Psychoeducation using a leaflet	26 weeks (Week 0, 13, 26, 52)	1. HADS; 2. PHQ-9; 3. GAD-7; 4. ISI; 5. MBI; 6. WHO-WPQ; 7. EQ-5D utility score
Ghawadra 2020 (Parallel RCT/Malaysia)	249→224 TG: 123→118 CG: 126→106	unclear	ICU, MU, SU, PED, OB & GYN	DASS-21 (stress (15–25), anxiety (8–14), depression (10–20))	Mindfulness-based training (initial 2 h workshop, and then self-practice via website)	No intervention	5 weeks (Week 0, 5, 8)	1. DASS-21; 2. JSS; 3. MAAS
Grabbe 2020 (Parallel RCT/America)	77→69 TG: 40→33 CG: 37→36	TG: 44.6 ± 13.4 (23–70) CG: 45.9 ± 13.0 (23–73)	ED, OR, ICU, specialty units, OPD, SU	NA	Community resiliency model (psychoeducation/sensory awareness skills training class, including mindful eating) (initial 3 h class, and then self-practice via application)	Nutrition/healthy eating (initial 3 h class, and then self-practice via application)	1 year (Week 0, 1, 12, 52)	1. WHO-5; 2. CDRS; 3. STSS; 4. CBI; 5. SSS-8
Zamanifar 2020 (Parallel RCT/Iran)	120→120 TG1: 30 TG2: 30 TG3: 30 CG: 30	TG1: 32.33 ± 4.59 TG2: 32.27 ± 4.66 TG3: 32 ± 5.53 CG: 32.60 ± 5.83	ED & PED-ED, ICU & PED-ICU, IM & PED-IM, PED-IU, NU, ONC, IM, NB	BAI ≥ 8	TG1: music therapy TG2: aromatherapy (with chamomile–lavender essential oil) TG3: aromatherapy & music therapy (20 min/session, 1 session/week)	No intervention	12 weeks (Week 0, 12)	BAI

Abbreviations. ANE, anesthesiology; BAI, Beck Anxiety Inventory; BJSQ, Brief Job Stress Questionnaire; BP, blood pressure; Brief-COPE, abbreviated version of the Coping Orientation to Problems Experienced Inventory; CBI, Copenhagen Burnout Inventory; CDRS, Connor-Davidson Resilience Scale; CG, control group; CICU, cardiac intensive care unit; DASS-21, Depression, Anxiety, and Stress Scales-21; ED, emergency department; EQ-5D, EuroQol five-dimension scale; FMI, Freiburg Mindfulness Inventory; FSS, Fatigue Severity Scale; GAD-7, Generalized Anxiety Disorder 7 Item Scale; GHQ, General Health Questionnaire; GYM, gynecologic; HADS, Hospital Anxiety and Depression Scale; HPLP-II, Health Promoting Lifestyle Profile II; HR, heart rate; ICU, intensive care unit; IM, internal medicine; IPD, inpatient ward; ISI, Insomnia Severity Index; IU, infectious unit; JSS, Job Satisfaction Scale; MAAS, Mindful Attention Awareness Scale; MBI, Maslach Burnout Inventory; MBSR, mindfulness-based stress reduction; MMSS, McCloskey/Mueller Satisfaction Scale; MU, medical unit; NA, not applicable; NB, new born room; NR, not reported; NSC, Nurse Stress Checklist; NSS, Nursing Stress Scale; NU, neonatal unit; OB, obstetric; ONC, oncology; OPD, outpatient department; OR, operation room; PANAS, Positive and Negative Affect Schedule; PDS, Posttraumatic Diagnostic Scale; PED, pediatric; PHQ-9, Patient Health Questionnaire-9; PILL, Pennebaker Inventory of Limbic Languidness; PKPCT v II, Power as Knowing Participation in Change Tool, Version II; PMR, progressive muscle relaxation; PSQI, Pittsburgh Sleep Quality Index; PSS, Perceived Stress Scale; PSY, psychiatric; QMWS, Questionnaire on Medical Workers’ Stress; RCT, randomized controlled trial; SAS, Self-Rating Anxiety Scale; SCL-90-R, Symptom Checklist-90-Revised; SDS, Self-Rating Depression Scale; SSS-8, Somatic Symptoms Scale-8; STAI, State-Trait Anxiety Inventory; STSS, Secondary Traumatic Stress Scale; SU, surgical unit; TG, treatment group; VAS, visual analogue scale; WHO-5, 5-item World Health Organization Well-Being Index; WHO-WPQ, World Health Organization Heath and Work Performance Questionnaire; WHOQOL-BREF, abbreviated World Health Organization Quality of Life questionnaire.

**Table 2 ijerph-18-08855-t002:** Main results of included studies.

Outcomes	Comparison (TG vs. CG)	Results	Reference
**Outcomes on occupation and environment**			
(1) JSS (job satisfaction)	MBIs vs. No intervention	By generalized estimating equations, Wald Chi–Square and df value (time x group interaction) was presented. Wald Chi-Square = 7.594 (2), df = 2 (*p* = 0.040)	Ghawadra 2020
(2) MMSS (job satisfaction)	MBSR vs. No intervention	TG: 102.27 ± 14.44, CG: 96.17 ± 18.05 (*p* > 0.05)	Lin 2019
(3) WHO-WPQ (presenteeism)	MBIs vs. Psychoeducation	(1) Absolute presenteeism -TG: 59.0 ± 14.19798, CG: 55.4 ± 13.55262 (*p* = 0.258) (2) Relative presenteeism -TG: 0.93 ± 0.322681, CG:0.88 ± 0.322681 (*p* = 0.065)	Watanabe 2019
**Outcomes on the individual’s resistance to stress**			
(1) Brief-COPE (coping style)	PMR vs. Education	Median value (25th–75th quartiles) (1) Use of instrumental support –TG: 5.5 (4.0–7.0), CG: 6.0 (5.0–6.0) (*p* = 0.980) (2) Humor -TG: 4.0 (3.0–4.7), CG: 4.0 (3.0–5.0) (*p* = 0.425) (3) Active coping -TG: 6.0 (6.0–7.0), CG: 6.0 (5.2–6.7) (*p* = 0.237) (4) Substance use -TG: 2.0 (2.0–2.0), CG: 2.0 (2.0–2.7) (*p* = 0.631) (5) Acceptance -TG: 6.0 (5.0–7.0), CG: 5.0 (4.0–6.0) (*p* = 0.038) (6) Venting -TG: 6.0 (5.0–6.0), CG: 5.0 (5.0–6.0) (*p* = 0.235) (7) Religion -TG: 6.0 (6.0–6.7), CG: 6.0 (4.2–6.0) (*p* = 0.108) (8) Denial: TG-3.5 (3.0–5.0), CG: 3.5 (2.2–4.0) (*p* = 0.302) (9) Behavioral disengagement –TG: 2.0 (2.0–3.0), CG: 3.0 (2.0–3.0) (*p* = 0.413) (10) Self-distraction -TG: 6.0 (6.0–7.0), CG: 6.0 (5.2–7.0) (*p* = 0.224) (11) Self-blame -TG: 4.0 (4.0–6.0), CG: 4.5 (4.0–5.0) (*p* = 0.758) (12) Positive reframing -TG: 6.0 (6.0–7.0), CG: 6.0 (6.0–7.0) (*p* = 0.295) (13) Use of emotional support -TG: 6.0 (5.0–7.0), CG: 5.0 (4.0–6.0) (*p* = 0.101) (14) Planning -TG: 6.0 (6.0–7.0), CG: 6.0 (5.0–7.0) (*p* = 0.160)	Ozgundondu 2019
(2) HPLP–II (healthy lifestyle)	Yoga vs. Usual care	TG: 3.08 ± 0.40, CG: 2.69 ± 0.38 (*p* = 0.006)	Alexander 2015
(3) CDRS (resilience)	Multimodal resilience training program vs. No intervention	Median value TG: 78, CG: 79 (The two groups were not statistically compared.)	Mealer 2014
	MBSR vs. No intervention	TG: 57.98 ± 11.58, CG: 55.11 ± 12.80 (p > 0.05)	Lin 2019
	Community resiliency model (including mindful eating) vs. Nutrition/healthy eating	TG: 31.72 ± 4.02, CG: 30.54 ± 4.99 (*p* = 0.910)	Grabbe 2020
(4) PKPCT v II (power)	Meditation vs. No intervention	(1) Global score –TG: 228.20 ± 36.97, CG:214.10 ± 33.82 (*p* = 0.001) (2) Awareness –TG:55.20 ± 9.60, CG:51.50 ± 8.44 (*p* = 0.049) (3) Choices –TG:56.80 ± 10.64, CG:52.90 ± 11.13 (*p* = 0.017) (4) Freedom –TG:59.70 ± 9.75, CG: 54.10 ± 10.67 (*p* = 0.005) (5) Involvement –TG: 56.60 ± 9.01, CG:54.60 ± 7.32 (*p* = 0.001)	Chang 2016
(5) FMI (mindfulness)	Yoga vs. Usual care	TG: 43.60 ± 7.32, CG: 39.65 ± 7.07 (*p* = 0.067)	Alexander 2015
(6) MAAS (mindfulness)	MBIs vs. No intervention	By generalized estimating equations, Wald Chi–Square and df value (time x group interaction) was presented. Wald Chi–Square = 0.066 (2), df = 2 (*p* = 0.967)	Ghawadra 2020
**Outcomes on the Individual’s Global Health and Wellness**			
(1) WHOQOL-BREF (quality of life)	Meditation vs. No intervention	(1) Global score -TG: 54.00 ± 9.20, CG: 52.50 ± 6.77 (*p* = 0.006) (2) Physical -TG: 13.90 ± 2.90, CG:13.40 ± 1.90 (*p* = 0.018) (3) Psychological -TG: 13.00 ± 2.96, CG: 13.30 ± 1.80 (*p* = 0.039) (4) Social -TG: 13.70 ± 2.63, CG: 12.60 ± 2.15 (*p* = 0.034) (5) Environmental -TG: 13.60 ± 2.05, CG: 13.20 ± 2.26 (*p* = 0.057)	Chang 2016
	Yoga vs. No intervention	(1) Global score -TG: 72.8 ± 2.8, CG: 62.4 ± 2.2 (*p* < 0.001) (2) Physical -TG: 70.14 ± 3.1, CG: 62.1 ± 2.3 (*p* < 0.001) (3) Psychological -TG: 73.3 ± 3.0, CG: 62.2 ± 1.8 (*p* < 0.001) (4) Social -TG: 72.6 ± 2.8, CG: 64.2 ± 2.4 (*p* < 0.001) (5) Environment -TG: 75.5 ± 2.4, CG: 61.3 ± 2.5 (*p* < 0.001)	Rostami 2019
(2) EQ-5D (quality of life)	MBIs vs. Psychoeducation	(1) Utility -TG: 0.85 ± 0.129073, CG: 0.88 ± 0.129073 (*p* = 0.131)	Watanabe 2019
(3) WHO-5 (well–being)	Community resiliency model (including mindful eating) vs. Nutrition/healthy eating	TG: 70.24 ± 16.74, CG: 62.46 ± 18.93 (*p* = 0.168)	Grabbe 2020
(4) GHQ (general health)	Relaxation vs. Traditional in-service education	Repeated measures ANOVA, interaction effect of treatment and time F [1, 132] = 1.86, *p* < 0.05	Tsai 1993
	Stretch-release relaxation vs. Cognitive relaxation vs. No intervention	TG1: 26.24 ± 9.23, TG2: 24.78 ± 7.03, CG: 28.83 ± 10.35 (*p* = 0.320)	Yung 2004
**Outcomes on Psychological Symptoms**			
(1) SCL-90-R (psychological pathology)	MBSR vs. Routine psychological support and activities	TG:119.6 ± 21.6, CG:132.6 ± 24.9 (*p* < 0.001)	Yang 2018
(2) MBI (burnout) - The primary outcome	Multimodal resilience training program vs. No intervention	Median value (25th-75th quartiles) (1) Emotional exhaustion -TG: 13.0 (8–28), CG: 25.0 (13–28) (2) Depersonalization - TG: 9 (5–16), CG: 10 (7–15) (3) Lack of personal accomplishment -TG: 37 (30–42), CG: 32 (28–40) (The two groups were not statistically compared.)	Mealer 2014
	Yoga vs. Usual care	(1) Emotional exhaustion -TG: 12.95 ± 8.76, CG: 20.60 ± 12.09 (*p* = 0.041) (2) Depersonalization -TG: 2.50 ± 3.65, CG: 5.15 ± 4.51 (*p* = 0.035) (3) Lack of personal accomplishment -TG: 39.60 ± 8.90, CG: 37.05 ± 9.98 (*p* = 0.554)	Alexander 2015
	MBI vs. Psychoeducation	(1) Emotional exhaustion -TG: 24.3 ± 9.35776, CG: 21.7 ± 8.712398 (*p* = 0.341) (2) Depersonalization -TG: 7.3 ± 4.840221, CG: 8.4 ± 4.840221 (*p* = 0.266) (3) Lack of personal accomplishment -TG: 22.2 ± 6.776309, CG: 22.2 ± 6.776309 (*p* = 0.664)	Watanabe 2019
(3) CBI (burnout) - The primary outcome	Community resiliency model (including mindful eating) vs. Nutrition/healthy eating	TG: 43.90 ± 18.32, CG: 38.22 ± 20.26 (*p* = 0.777)	Grabbe 2020
(4) NSC (stress)	Relaxation vs. Traditional in–service education	Repeated measures ANOVA, interaction effect of treatment and time F [1, 132] = 12.5, *p* < 0.05	Tsai 1993
(5) NSS (stress)	MBSR vs. Routine psychological support and activities	TG: 68.2 ± 9.1, CG: 83.1 ± 8.4 (*p* < 0.001)	Yang 2018
(6) Self-perceived stress (stress)	Music vs. Chair rest	TP: 2.98 ± 1.51, CP: 4.78 ± 1.62 (*p* < 0.001)	Lai 2011
(7) PSS (stress)	MBSR vs. No intervention	TG: 37.39 ± 5.97, CG: 40.76 ± 5.01 (*p* < 0.01)	Lin 2019
	PMR vs. Education	Median value (25th–75th quartiles) TG: 27.00 (25.00–29.75), CG: 29.00 (27.00–31.75) (*p* = 0.030)	Ozgundondu 2019
(8) BJSQ (stress)	Yoga vs. Stress relief method	TP: 56.1 ± 8.5, CP: 64.1 ± 12.7 (*p* = 0.01)	Miyoshi 2019
(9) QMWS (stress)	Yoga vs. No intervention	Number of QMWS score > 32 (high stress) TG: 19/54, CG: 39/51 (*p* = 0.001)	Fang 2015
(10) STSS (posttraumatic stress)	Community resiliency model (including mindful eating) vs. Nutrition/healthy eating	TG: 32.31 ± 9.53, CG: 30.30 ± 9.56 (*p* = 0.846)	Grabbe 2020
(11) PDS (posttraumatic stress)	Multimodal resilience training program vs. No intervention	Median value (25th–75th quartiles) TG: 37 (30–42), CG: 32 (28–40) (The two groups were not statistically compared.)	Mealer 2014
(12) STAI-state (anxiety state)	Stretch-release relaxation vs. Cognitive relaxation vs. No intervention	TG1: 38.35 ± 7.36, TG2: 36.89 ± 5.75, CG: 41.48 ± 8.16 (*p* = 0.097)	Yung 2004
(13) STAI-trait (anxiety trait)	Stretch-release relaxation vs. Cognitive relaxation vs. No intervention	TG1: 43.59 ± 6.58, TG2: 42.06 ± 6.26, CG: 40.48 ± 6.33 (*p* = 0.679)	Yung 2004
(14) DASS-21 (depression, anxiety, stress)	MBIs vs. No intervention	By generalized estimating equations, Wald Chi–Square and df value (time × group interaction) was presented. (1) Stress -Wald Chi–Square = 3.673 (2), df = 2 (*p* = 0.159) (2) Anxiety -Wald Chi–Square = 9.694 (2), df = 2 (*p* = 0.008) (3) Depression -Wald Chi–Square = 0.686 (2), df = 2 (*p* = 0.709)	Ghawadra 2020
(15) PANAS (positive and negative emotion)	MBSR vs. No intervention	(1) Positive emotion/affect –TG: 32.02 ± 6.45, CG: 29.00 ± 5.51 (*p* < 0.05) (2) Negative emotion/affect –TG: 20.80 ± 4.72, CG: 23.61 ± 5.17 (*p* < 0.01)	Lin 2019
(16) HADS (depression, anxiety)	Multimodal resilience training program vs. No intervention	Median value (25th–75th quartiles) (1) Anxiety -TG: 12.0 (10–13), CG: 11 (10–12) (2) Depression - TG: 9.0 (7–10), CG: 9.0 (8–11) (The two groups were not statistically compared.)	Mealer 2014
	Percussion music vs. No intervention	(1) Depression -TG: 13.23 ± 2.83, CG: 13.04 ± 2.72 (*p* > 0.05)	Ploukou 2018
	MBIs vs. Psychoeducation	(1) Depression -TG: 3.21 ± 2.25877, CG: 2.54 ± 2.226502 (*p* = 0.192) (2) Anxiety -TG: 3.98 ± 2.226502, CG: 3.43 ± 2.161965 (*p* = 0.190)	Watanabe 2019
(17) SAS (anxiety)	MBSR vs. Routine psychological support and activities	TG:36.4 ± 7.1, CG: 45.1 ± 6.7 (*p* < 0.001)	Yang 2018
(18) BAI (anxiety)	Music therapy vs. Aromatherapy vs. Music & Aromatherapy vs. No intervention	TG1: 39.74 ± 8.45, TG2: 37.83 ± 8.79, TG3: 39.97 ± 9.38, CG: 51.37 ± 9.58 (*p* = 0.999 for TG1 vs. TG2; *p* = 0.999 for TG2 vs. TG3; *p* = 0.0001 for TG2 vs. CG; *p* = 0.999 for TG1 vs. TG3; *p* = 0.0001 for TG1 vs. CG; *p* = 0.0001 for TG3 vs. CG)	Zamanifar 2020
(19) GAD-7 (anxiety)	MBIs vs. Psychoeducation	TG: 4.13 ± 2.484647, CG: 3.11 ± 2.387842 (*p* = 0.057)	Watanabe 2019
(20) SDS (depression)	MBSR vs. Routine psychological support and activities	TG: 35.4 ± 8.3, CG: 41.2 ± 8.7 (*p* < 0.001)	Yang 2018
(21) PHQ-9 (depression)	MBIs vs. Psychoeducation	TG: 5.78 ± 3.065473, CG: 4.97 ± 2.936401 (*p* = 0.315)	Watanabe 2019
(22) PSQI (insomnia)	Yoga vs. No intervention	(1) Global score –TG: 7.61 ± 1.25, CG: 10.31 ± 2.42 (2) Sleep quality –TG: 1.34 ± 0.35, CG: 1.68 ± 0.31 (3) Sleep duration –TG: 1.34 ± 0.09, CG: 1.68 ± 0.45 (4) Sleep efficiency –TG: 1.42 ± 0.11, CG: 1.79 ± 0.38 (5) Sleep disturbance –TG: 1.51 ± 0.17, CG: 1.93 ± 0.45 (6) Use of sleep medication –TG: 1.41 ± 0.23, CG: 1.79 ± 0.34 (7) Daytime dysfunction –TG: 1.24 ± 0.11, CG: 1.83 ± 0.41 (The two groups were not statistically compared.)	Fang 2015
(23) ISI (insomnia)	MBIs vs. Psychoeducation	TG: 6.18 ± 3.743104, CG: 5.35 ± 3.549495 (*p* = 0.435)	Watanabe 2019
**Outcomes on Somatic Symptoms**			
(1) SSS-8 (somatic symptom)	Community resiliency model (including mindful eating) vs. Nutrition/healthy eating	TG: 5.81 ± 4.55, CG: 5.27 ± 4.26 (*p* = 0.563)	Grabbe 2020
(2) PILL (psychosomatic symptom)	Percussion music vs. No intervention	Median value (25th–75th quartiles) TG: 99.5 (77.75–128.25), CG: 114.5 (86.75–144.75) (*p* > 0.05)	Ploukou 2018
(3) FSS (fatigue)	PMR vs. Education	TG: 30.86 ± 10.41, CG: 42.82 ± 9.66 (*p* < 0.001)	Ozgundondu 2019
**Outcomes on Biological Data**			
(1) mean arterial BP (mmHg)	Music vs. Chair rest	TP: 84.80 ± 7.54, CP: 90.52 ± 7.75 (*p* < 0.001)	Lai 2011
(2) serum cortisol (nmol/mmol)	Music vs. Chair rest	TP: 4.97 ± 3.42, CP: 6.42 ± 3.46 (*p* < 0.025)	Lai 2011
(3) HR (per minute)	Music vs. Chair rest	TP: 65.44 ± 8.82, CP: 69.06 ± 9.55 (*p* < 0.001)	Lai 2011
(4) finger temperature (℃)	Music vs. Chair rest	TP: 26.92 ± 4.70, CP: 24.11 ± 4.53 (*p* < 0.001)	Lai 2011

Abbreviations: ANOVA, analysis of variance; BAI, Beck Anxiety Inventory; BJSQ, Brief Job Stress Questionnaire; BP, blood pressure; Brief-COPE, abbreviated version of the Coping Orientation to Problems Experienced Inventory; CBI, Copenhagen Burnout Inventory; CDRS, Connor-Davidson Resilience Scale; CG, control group; CP, control period; DASS-21, Depression, Anxiety, and Stress Scales-21; EQ-5D, EuroQol five-dimension scale; FMI, Freiburg Mindfulness Inventory; FSS, Fatigue Severity Scale; GAD-7, Generalized Anxiety Disorder 7 Item Scale; GHQ, General Health Questionnaire; HADS, Hospital Anxiety and Depression Scale; HPLP-II, Health Promoting Lifestyle Profile II; HR, heart rate; ISI, Insomnia Severity Index; JSS, Job Satisfaction Scale; MAAS, Mindful Attention Awareness Scale; MBI, Maslach Burnout Inventory; MBIs, mindfulness-based interventions; MBSR, mindfulness-based stress reduction; MMSS, McCloskey/Mueller Satisfaction Scale; NSC, Nurse Stress Checklist; NSS, Nursing Stress Scale; PANAS, Positive and Negative Affect Schedule; PDS, Posttraumatic Diagnostic Scale; PHQ-9, Patient Health Questionnaire-9; PILL, Pennebaker Inventory of Limbic Languidness; PKPCT v II, Power as Knowing Participation in Change Tool, Version II; PMR, progressive muscle relaxation; PSQI, Pittsburgh Sleep Quality Index; PSS, Perceived Stress Scale; QMWS, Questionnaire on Medical Workers’ Stress; SAS, Self-Rating Anxiety Scale; SCL-90-R, Symptom Checklist-90-Revised; SDS, Self-Rating Depression Scale; SSS-8, Somatic Symptoms Scale-8; STAI, State-Trait Anxiety Inventory; STSS, Secondary Traumatic Stress Scale; TG, treatment group; TP, treatment period; WHO-5, 5-item World Health Organization Well-Being Index; WHO-WPQ, World Health Organization Heath and Work Performance Questionnaire; WHOQOL-BREF, abbreviated World Health Organization Quality of Life questionnaire.

## Data Availability

These data used to support the findings of this study are included within the article.

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
