# Peer review of "The Effectiveness and Safety of Mind-Body Modalities for Mental Health of Nurses in Hospital Setting: A Systematic Review"

_ijerph, 2021, doi:10.3390/ijerph18168855_

Round 1

Reviewer 1 Report

  • I believe that the article would benefit from a breakdown of what the authors include when they talk about mental health problems. It is not until the reader get to the results section that this becomes clear, it would be better to be upfront about this early on. Greater specificity would add further appeal of the study and easier for readers to see the applicability of the findings in their own context. 
  • In relation to the study search further articles could potentially be found by canvasing the reference lists of included sources. If this was a methodology used then it should be described in the search section. 
  • Unclear whether only English language sources were selected for review? This should be clarified to ascertain the scope of the review.
  • It is concerning that the authors have chosen to use the Cochrane Risk of Bias tool and then not being able to assess publication bias (R127-129), which defeat the purpose of choosing this tool. If they were unable to assess bias with this tool other parameters could and should have been used to assess bias and quality of chosen studies, as it stands it is impossible to determine the quality of chosen articles, which raises question marks as to the outcome of the authors systematic review. 

Author Response

  • Response to Comments from Reviewer 1

Comment 1:

I believe that the article would benefit from a breakdown of what the authors include when they talk about mental health problems. It is not until the reader get to the results section that this becomes clear, it would be better to be upfront about this early on. Greater specificity would add further appeal of the study and easier for readers to see the applicability of the findings in their own context.  

Response:              

Thank you for the comment. We think this comment pointed out that there was no detailed explanation of the rationale and purpose of this review in the Introduction section, and we agree with that comment. Therefore, we have enhanced the Introduction section by adding more sentences in the revised manuscript to give readers an idea of what they can get from this review.

“Likewise, mind-body modalities have the potential to improve nurses’ overall health and level of well-being and prevent and/or reduce burnout levels by alleviating the accompanying physical symptoms as well as improving the psychological stress of nurses. For example, a tertiary care hospital in the United States reported that mindfulness-based stress reduction (MBSR), a typical type of mindfulness-based interventions (MBIs), was introduced to staff nurses in the hospital, and it was reported that the program effectively reduced job burnout and improved mindfulness, self-compassion, and serenity in the participants [14]. Mindfulness practice is also being considered as an effective self-management method or a stress reduction technique for healthcare workers including nurse exposed to the COVID-19 pandemic and threatened with their mental health and well-being [15,16]. Therefore, examining the impact of mind-body modalities on the mental health of nurses will potentially help establish strategies to improve the mental health of healthcare workers in this unprecedented pandemic, and further potentially increase humanity's capacity to respond to this pandemic. However, no study has systematically analyzed the effectiveness of mind-body modalities on the mental health problems of nurses. The purpose of this systematic review was hence to evaluate whether mind-body modalities improve burnout and other mental health aspects of nurses in hospital setting.” (Please see in page 2, red words)

References

  1. Penque, S. Mindfulness to promote nurses' well-being. Nursing management 2019, 50, 38-44, doi:10.1097/01.NUMA.0000557621.42684.c4.
  2. Søvold, L.E.; Naslund, J.A.; Kousoulis, A.A.; Saxena, S.; Qoronfleh, M.W.; Grobler, C.; Münter, L. Prioritizing the Mental Health and Well-Being of Healthcare Workers: An Urgent Global Public Health Priority. Front Public Health 2021, 9, 679397-679397, doi:10.3389/fpubh.2021.679397.
  3. Callus, E.; Bassola, B.; Fiolo, V.; Bertoldo, E.G.; Pagliuca, S.; Lusignani, M. Stress Reduction Techniques for Health Care Providers Dealing With Severe Coronavirus Infections (SARS, MERS, and COVID-19): A Rapid Review. Front Psychol 2020, 11, 589698-589698, doi:10.3389/fpsyg.2020.589698.

Comment 2:

In relation to the study search further articles could potentially be found by canvasing the reference lists of included sources. If this was a methodology used then it should be described in the search section.

Response:              

Thank you for the comment. We have described in the original manuscript that we have reviewed a list of references from related papers including the studies included in this review with the sentence “In addition, a manual search on Google Scholar was conducted to search for gray and potentially missing literature, and a list of references from related papers was reviewed accordingly.”. But we think the expression was not clear. Therefore, we modified the sentence:

“In addition, a manual search on Google Scholar was conducted to search for gray and potentially missing literature, and a list of references from related papers including the studies included in this review was reviewed accordingly.” (Please see in page 2, red words)

Comment 3:

Unclear whether only English language sources were selected for review? This should be clarified to ascertain the scope of the review.

Response:              

Thank you for the comment. We have described in the original manuscript that there are no restrictions on language sources in the selection for review with the sentence “Randomized controlled trials (RCTs) were conducted, no restrictions on language were imposed in the study”. But we think the expression alone was not enough. Therefore, we added one more sentence:

“Randomized controlled trials (RCTs) were conducted, and no restrictions on language were imposed in the study. That is, there are no restrictions on language sources in the selection for review.” (Please see in page 3, red words)

Comment 4:

It is concerning that the authors have chosen to use the Cochrane Risk of Bias tool and then not being able to assess publication bias (R127-129), which defeat the purpose of choosing this tool. If they were unable to assess bias with this tool other parameters could and should have been used to assess bias and quality of chosen studies, as it stands it is impossible to determine the quality of chosen articles, which raises question marks as to the outcome of the authors systematic review.

Response:              

Thank you for the comment. Our response to this comment should be premised on the following explanation. As described in the manuscript, this study complies with the PRISMA statement, the standard reporting system of systematic review methodology.

“This systematic review complied with the Preferred Reporting Items for Systematic Reviews and Meta-Analyses (PRISMA) statement for reporting systematic reviews and meta-analyses [17] (Supplementary File 1). The protocol of this systematic review was registered in OSF registries (doi: 10.17605/OSF.IO/U8P3T), and this review followed the protocol.” (Please see in page 2)

As described in item #14 of the PRISMA statement, reporting bias or publication bias can be assessed through the results of synthesis, i.e. meta-analysis. In this case, some additional methods such as generating funnel plots can be used. However, statistical analysis methods that allow analyze publication bias in systematic review without meta-analysis are not yet established.

“Due to the heterogeneity of the interventions and outcomes of the included studies, quantitative synthesis was not performed in this study. Therefore, publication bias using a funnel plot could not be evaluated.” (Please see in page 3)

The Cochrane Risk of Bias tool is a tool that enables evaluation of the methodological quality of each included RCT, but it is not a tool that allows evaluation of publication bias. Therefore, the Cochrane Risk of Bias tool is not related to publication bias.

“This tool judges the RoB of RCTs as high, low, or unclear in the domains of selection, performance, detection, attrition, reporting, and other biases.” (Please see in page 3)

In summary, publication bias cannot be statistically analyzed in our systematic review (without meta-analysis), and this problem may be not solved even if a tool other than the Cochrane Risk of Bias tool is used. However, we think that the inability to measure publication bias may be one of the limitations of this systematic review. Therefore, in the revised manuscript, we added a limitation regarding evaluation of publication bias in the Discussion section as follows.

Finally, because quantitative synthesis was not performed due to the heterogeneity between the studies included in this review, evaluation of publication bias through funnel plot generation was not possible [60], but this does not prove the absence of publication bias. That is, the possibility of publication bias suggests that unpublished studies in this field may have exaggerated the net benefit of the intervention [61]. Therefore, the findings of this review should be interpreted with caution.” (Please see in pages 21-22, red words)

References

  1. Parekh-Bhurke, S.; Kwok, C.S.; Pang, C.; Hooper, L.; Loke, Y.K.; Ryder, J.J.; Sutton, A.J.; Hing, C.B.; Harvey, I.; Song, F. Uptake of methods to deal with publication bias in systematic reviews has increased over time, but there is still much scope for improvement. Journal of clinical epidemiology 2011, 64, 349-357, doi:10.1016/j.jclinepi.2010.04.022.
  2. Murad, M.H.; Chu, H.; Lin, L.; Wang, Z. The effect of publication bias magnitude and direction on the certainty in evidence. BMJ Evid Based Med 2018, 23, 84-86, doi:10.1136/bmjebm-2018-110891.

Reviewer 2 Report

Thank you for the opportunity to review this paper. the paper sought to review the effectiveness and safety of mind-body modalities for mental health of nurses in hospital setting. 

The work has merit, is well written and structured. Only one minor point is needed to write clearly.

Line 124: p-value less than 0.05 was considered to be statistically significant. However, the data has been reported significant improved quality of life (p=0.006 to 0.057) in line 260-263. 

Author Response

  • Response to Comments from Reviewer 2

Comment 1:

Thank you for the opportunity to review this paper. the paper sought to review the effectiveness and safety of mind-body modalities for mental health of nurses in hospital setting. The work has merit, is well written and structured. Only one minor point is needed to write clearly.

Response:              

Thank you for your valuable comments on our manuscript.

Comment 2:

Line 124: p-value less than 0.05 was considered to be statistically significant. However, the data has been reported significant improved quality of life (p=0.006 to 0.057) in line 260-263.

Response:              

Thank you for the comment. In this revised manuscript, that sentence has been corrected as follows:

“Regarding the outcomes of the individual’s global health and wellness, Chang (2016) [48] found that, compared to no intervention, meditation significantly improved global score and most subscales of quality of life assessed using the abbreviated World Health Organization Quality of Life (WHOQOL-BREF) questionnaire (p = 0.006 to 0.039 for global score, physical domain, psychological domain, and social domain; p = 0.057 for environmental domain), while Watanabe (2019) [55] found no significant difference in quality of life assessed by the EuroQol five-dimension scale utility score between the MBIs group and the psychoeducation group (p = 0.131).” (Please see in page 18, red words)

Reviewer 3 Report

As the authors pointed out, no study was found to evaluate the effectiveness of mind-body modalities on the mental health problems of nurses by systematic reviews.  Taking the mental health problems of nurses such as burnout, this kind of evaluation is important.  However, the authors are required to answer the following reviewer's identification.

The readers can easily understand the 17 studies' characteristics and results by table 1 and 2, and the risk of bias in figure 2.  But the reviewer is afraid that the table 3 mislead and misunderstand the interpretation of this review because the numbers of studies included in this review is small pointed out by the authors themselves in discussion.  The authors should reconsider the table 3 use in this manuscript. 

Author Response

  • Response to Comments from Reviewer 3

Comment 1:

As the authors pointed out, no study was found to evaluate the effectiveness of mind-body modalities on the mental health problems of nurses by systematic reviews.  Taking the mental health problems of nurses such as burnout, this kind of evaluation is important.  However, the authors are required to answer the following reviewer's identification.

Response:              

Thank you for your valuable comments on our manuscript.

Comment 2:

The readers can easily understand the 17 studies' characteristics and results by table 1 and 2, and the risk of bias in figure 2.  But the reviewer is afraid that the table 3 mislead and misunderstand the interpretation of this review because the numbers of studies included in this review is small pointed out by the authors themselves in discussion.  The authors should reconsider the table 3 use in this manuscript.

Response:              

Thank you for the comment. The reason we summarized the results in Table 3 in the original manuscript was to improve the readability of the readers. However, as the reviewer commented, we agree that this, despite the lack of included studies, can be misleading as if presenting definitive results. Therefore, we deleted the table 3 from the revised manuscript.

Round 2

Reviewer 3 Report

The authors properly revised the manuscript.